



# A novel explainable deep learning framework for reconstructing South Asian palaeomonsoons

Kieran M. R. Hunt[1,2] and Sandy P. Harrison[3]

[1]Department of Meteorology, University of Reading, Reading, UK
[2]National Centre for Atmospheric Sciences, University of Reading, Reading UK
[3]Department of Geography, University of Reading, Reading, UK

**Correspondence:** Kieran M. R. Hunt (k.m.r.hunt@reading.ac.uk)

**Abstract.**

We present novel explainable deep learning techniques for reconstructing South Asian palaeomonsoon rainfall over the last 500 years, leveraging long instrumental precipitation records and palaeoenvironmental datasets from South and East Asia to build two types of model: dense neural networks ('timeline models') and convolutional neural networks (CNNs). The

timeline models are trained individually on seven regional rainfall datasets and while they capture decadal-scale variability and significant droughts, they underestimate interannual variability. The CNNs, designed to account for spatial relationships in both predictor and target, demonstrate higher skill in reconstructing rainfall patterns and produce robust spatiotemporal reconstructions. The 19th and 20th centuries were characterised by marked inter-annual variability in the monsoon, but earlier periods were characterised by more decadal- to centennial-scale oscillations. Multidecadal droughts occurred in the mid-

seventeenth and nineteenth centuries, while much of the eighteenth century (particularly the early part of the century) was characterised by above-average monsoon precipitation. Extreme droughts tend to be concentrated in south and west India and often coincide with recorded famines. By applying explainability techniques, we show that the models make use of both local hydroclimate and synoptic-scale dynamical relationships. Our findings offer insights into the historical variability of the Indian summer monsoon and highlight the potential of deep learning techniques in palaeoclimate reconstruction.

## 1   Introduction

### 1.1   The Indian summer monsoon

The Indian, or South Asian, summer monsoon occurs each year between June and September. It brings about 80% of annual rainfall to the subcontinent, supporting the lives and livelihoods of over a billion people (Turner and Annamalai, 2012). While the Indian monsoon can be thought of as a large-scale convectively coupled land-sea breeze, there is strong hetereogeneity in

both space and time which is forced through both internal and external variability (Rind and Overpeck, 1993; Webster et al., 1998).

The interannual variability of the monsoon, averaged over the whole of India, is slight – one standard deviation amounts to only about 10% (Webster et al., 1998). Yet intraseasonal variability, mostly forced by the Boreal Summer Intraseasonal





Oscillation (BSISO), can be many times larger (Kikuchi et al., 2012; Kikuchi, 2021; Hunt and Turner, 2022). This can lead to

extended periods of substantially below or above average rainfall (Krishnamurthy and Shukla, 2000; Goswami and Ajayamohan, 2001; Pai et al., 2016); such as the month-long break of August 2023, when the whole country received only 64% of its typical August rainfall; or the twin low pressure systems (LPSs) in August 2018 that resulted in Kerala receiving more than twice the typical rainfall for the first three weeks of the month (Hunt and Menon, 2019).

Monsoon heterogeneity is also forced externally, through the multiple teleconnections or large-scale modes of variability

across the tropics and extratropics, identified both in palaeo- and present-day climates. These modes are not strictly orthogonal, nor are they mutually exclusive, and the strength of the teleconnections can vary on multidecadal timescales. Large-scale forcing and teleconnections, however, explain the majority of interannual variance in the summer monsoon. These include, but are not limited to: ENSO (Webster and Yang, 1992; Torrence and Webster, 1999; Turner et al., 2005; Xavier et al., 2007); the Indian Ocean Dipole (IOD; Ashok et al., 2001; Cherchi et al., 2021); solar forcing (Rind and Overpeck, 1993; Agnihotri

et al., 2002) and related latitudinal variations in the ITCZ (Fleitmann et al., 2007); as well as multidecadal variability in the Atlantic (Gupta et al., 2003; Archer and Fowler, 2004; Wang et al., 2005), Pacific (Krishnan and Sugi, 2003; Krishnamurthy and Krishnamurthy, 2014), and West African monsoon (Crétat et al., 2020, 2024).

## 1.2    Palaeoclimate reconstructions of the Indian monsoon

Robust palaeoclimate reconstructions of the Indian summer monsoon are challenging because of the small number of datasets

from the Indian peninsula (Wang et al., 2010; Rehfeld et al., 2013; Dixit and Tandon, 2016), regional variability within the monsoon (Ramesh and Yadava, 2005; Banerji et al., 2020), and strong interannual variability (Ramesh and Yadava, 2005; Dimri et al., 2022). These problems are further compounded because the impact of the monsoon on humans is predominantly felt on seasonal timescales but there are few palaeoclimate records with sufficient resolution to detect such a signal. Tree rings and speleothems can have sub-annual or annual resolution, but there are few such records available for India.

Tree rings, speleothems, and palaeoclimate records with lower resolution, such as lake and marine deposits, have proved useful in constraining changes in decadal variability. Studies using such records (e.g., Burns et al., 2002; Anderson et al., 2010; Dixit and Tandon, 2016; Kaushal et al., 2018; Banerji et al., 2020; Rawat et al., 2021) have shown a consistent and coherent strengthening of the summer monsoon during the Medieval Warm Period (MWP; c.950–1250 CE) and a weakening during the Little Ice Age (LIA; c. 1500–1850 CE). These responses are generally understood to be caused by shifts in the ITCZ in response

to changing solar forcing (Haug et al., 2001), but it is unclear if this is a local response (Sinha et al., 2011) or driven by ENSO (Burns et al., 2002). The summer monsoon has also experienced periods of extended drought (Sinha et al., 2011) and deluge (Sridhar and Chamyal, 2018) outside the MWP and LIA. These were mostly regional in nature, however, and thus do not always correlate well with indices such as all-India rainfall records. Furthermore, the interpretation of palaeoenvironmental records in terms of monsoon changes is complicated because the records can be influenced by changes specific to the depositional

setting, local hydrological conditions, regional signals, as well as changes in the relative importance of monsoonal and nonmonsoonal sources (Dixit and Tandon, 2016; Wolf et al., 2023). Nevertheless, longer term changes in monsoon strength during the Holocene have been reconstructed from speleothem records (Kaushal et al., 2018), specifically increased monsoon intensity





during the early Holocene (12–6 ka) with a gradual decrease in rainfall from the mid-Holocene to present day. Variability was higher in the late Holocene (after c. 5.6 ka) due to changes in solar irradiance, ENSO, and the PDO.

The history of the monsoon over the last two centuries has been reconstructed at annual or near-annual resolution using tree rings from fir and spruce across the Himalaya (Sano et al., 2012, 2017; Brunello et al., 2019; Sano et al., 2020; Fan et al., 2022; Thomte et al., 2022; Dhyani et al., 2023). These records show a general increase in aridity in the past two centuries and increased variability in the last few decades. However, the reconstructed precipitation records do not correlate well with each other, in part because of the impact of local conditions and inter-specific differences on tree growth but also reflecting regional

variations in rainfall across the Himalaya. A 500-year tree ring record from southern India (Borgaonkar et al., 2010), which correlates well with all-India rainfall over the instrumental record and has therefore been interpreted as a monsoon signal, showed no evidence of the increase in aridity over the last two centuries indicated by the Himalayan records. Instead, it has been interpreted as indicating weak monsoons with high variability from 1750–1850, followed by a period of strong monsoons at the end of the nineteenth century.

Cook et al. (2010) combined these various lines of information, using a point-by-point regression on a network of tree rings, including three from peninsular India, to construct a gridded 'Monsoon Asia Drought Atlas', spanning 1300–2005 CE at a resolution of 2.5°. They identified major Indian droughts in 1756–1758 (the 'Strange Parallels' drought), 1790–1796 (the 'East India' drought), and 1876–1878 (the 'Great Drought'). They also found that the monsoon was sensitive to different flavours of ENSO (i.e., whether the strongest anomalies are in the central Pacific or east Pacific). Very long instrumental records, some

of which stretch back to 1813 (Sontakke and Singh, 1996; Sontakke et al., 2008), indicate a period of weak monsoons in the middle of the nineteenth century.

  Most monsoon palaeoclimate studies have focused on single or small numbers of records. Even where they have used wider networks (e.g. Cook et al., 2010), they have not quantified or used the spatial relationships between the palaeoclimate records or within the rainfall field, so regional variations in monsoon rainfall are not well documented or understood. Machine learning

methods that can leverage this kind of information, even implicitly, could therefore be useful to reconstruct maps of monsoon evolution.

### 1.3 Data-driven approaches to reconstruct the palaeoclimate

Machine learning approaches have been used in palaeoclimate research for automated palaeoenvironmental record generation, model post-processing, and reconstruction. Automated palaeoenvironmental record generation has typically relied on image

detection and classification to improve the efficiency and accuracy of extracting chronologies, including layer counting in speleothems (Sliwinski et al., 2023) and tree ring width detection (e.g. Fabijańska and Danek, 2018; Kim et al., 2023; Poláček et al., 2023; Wu et al., 2023), as well as for pollen identification (e.g. Tcheng et al., 2016; de Geus et al., 2019; Bourel et al., 2020; Olsson et al., 2021), measuring soil organic carbon content (Lukens et al., 2019; Liu et al., 2022), measuring carbonate content in marine sediment (Lee et al., 2022), and identifying leaf physiognomy (Wei et al., 2021). Machine learning has also

been used to create backward models, for example, estimating tree ring width chronologies from local environmental factors



(Jevšenak et al., 2018; Bodesheim et al., 2022; Li et al., 2023). Nelson et al. (2021) also used machine learning to improve and extend instrumental records.

Machine learning has not been as widely used for model post-processing, although it has been used to improve the temporal resolution of model output using frame interpolation methods (Zheng et al., 2024), to reconstruct output variables through nonlinear mappings (Huang et al., 2020), for anomaly detection (Bianchette et al., 2023), and for identifying droughts (Coats et al., 2020). The use of machine learning for palaeoclimate reconstruction is a relatively unexplored field, although Bayesian machine learning methods are potentially more useful for this than simple linear regression techniques (e.g. Mannila et al., 1998; Chandra et al., 2021; Andermann et al., 2022). Neural network based methods, however, were used byMalmgren and Nordlund (1997) to reconstruct summer and winter SSTs over the southern Indian Ocean using a simple multilayer perceptron (MLP). Cortese et al. (2005) used an MLP to to estimate North Atlantic summer SSTs from protozoan assemblages over the Holocene. Guiot et al. (2005) used a two-layer MLP with 47 palaeoclimate records to reconstruct temperature and temperature-related events (e.g., grape harvests) in Western Europe. Carro-Calvo et al. (2013) used a simple MLP to reconstruct winter precipitation in Mediterranean Europe from 1700 CE onwards.

## 1.4 This study

In this study, we capitalise on the fact that India has some of the longest instrumental rainfall records in the world to test whether deep learning techniques can take advantage of the nonlinear relationships between the palaeoenvironmental records and seasonal rainfall, and between the palaeoenvironmental records themselves, to overcome the challenges raised by the sparsity of the palaeoclimate records and regional variation in the monsoon, and thus generate robust spatiotemporal reconstructions of the Indian summer monsoon over the past five centuries.

We first address three methodological questions:

1. How well does a simple multilayer perceptron model timeseries of regional monsoon rainfall anomalies?

2. Can a convolutional neural network take advantage of the spatial structures within these rainfall anomalies to improve spatial reconstructions beyond commonly used statistical methods?

3. How do the models combine information from the palaeoenvironmental records to build their reconstructions?

To answer these questions, we must first overcome the issue of having a comparatively small training dataset (Sec. 2): ~150 years for the timeline models and only 100 years for the CNN model. We use a variety of techniques to stabilise model training (Sec. 3). We then show that both the regional timeline models (Sec. 4.1) and CNN model (Sec. 4.2) produce robust and stable estimates of monsoon rainfall over the last 500 years. We compare these results to standard statistical models that have often been applied for reconstructions and explore the implications of our approach. We then apply explainable AI methods to our models (Sec. 4.3) and analyse the different kinds of contributions made by individual palaeoclimate records to the final predictions. We then describe the history of the Indian monsoon over the past 500 years (Sec. 4.4) before discussing the implications (Sec. 5) and conclusions of our work (Sec. 6).





**Figure 1.** Locations of the 155 palaeoclimate proxy records used in this study, coloured by type. Blue contours denote isohyets of the summer monsoon (June–September 1940–2022, computed using ERA5) and grey shading denotes orographic height. Also shown, in red, are the seven homogeneous rainfall regions of India (Sontakke et al., 2008).

## 2 Data

### 2.1 Palaeoclimate records

#### 2.1.1 PAGES2k global 2,000 year multiproxy temperature database v2.0.0

The PAGES2k dataset (PAGES2k Consortium, 2017) is a collection of temperature reconstructions for the Common Era (1 CE to present) designed to contextualise industrial-era warming within natural climatic variability. Version 2.0.0 of the PAGES2k



temperature database augments the earlier PAGES2k-2013 collection of terrestrial records with marine records. It also includes more terrestrial records, new metadata, and improved validation, making the dataset more cohesive and uniformly structured across regions. The dataset comprises 692 records from 648 locations covering all continental regions and major ocean basins. These records, sourced from trees, ice, sediment, corals, speleothems, documentary evidence, and other archives, range in length from 50 to 2000 years with a median length of 547 years. The temporal resolution varies from biweekly to centennial. Almost half of the palaeoclimate time series in this dataset correlate significantly with the HadCRUT4.2 surface temperature (Morice et al., 2012) over the period 1850–2014. The global temperature composites derived from these data show good coherence between high- and low-resolution records across different archive types and geographical locations.

### 2.1.2 Iso2k global palaeo water isotope database v1.0.0

The Iso2k database (Konecky et al., 2020) is a collection of stable oxygen ($\delta^{18}$O) and hydrogen ($\delta^2$H) isotope records from precipitation, seawater, lake water, soil, and groundwater, reflecting hydroclimate changes over the Common Era. The isotope records are compiled from a variety of natural archives including glaciers, ground ice, cave formations, corals, sclerosponges, mollusc shells, and wood. The database comprises 759 isotope records with individual datasets having a temporal resolution ranging from sub-annual to centennial.

### 2.1.3 Northern Hemisphere hydroclimatic variability database v1.0.0

The Northern Hemisphere hydroclimatic variability database (Ljungqvist et al., 2016) comprises 196 hydroclimate reconstructions and 128 temperature reconstructions, with a minimum length of 1000 years, from sources including tree rings, speleothems, and sediments. The temporal resolution varies from annual to multicentennial.

### 2.1.4 Homogenisation

The three databases were combined, homogenised, and filtered, as follows:

1. All .lpd (LiPD) and .txt files in the relevant directories were identified. Each one corresponds to a single palaeoenvironmental record. These were read in using the *lipd* and *pandas* python libraries.

2. Datasets missing a 'year' attribute were discarded, as were those that do not extend into the period after 1900 CE, those that have year values after 2020 CE, and those whose year attribute is non-monotonic.

3. The remaining datasets were checked for duplicates, which were removed. Entries were considered duplicates if both the year and record value matched one another.

4. Datasets with fewer than three entries were also removed.

5. Datasets were then rebased onto a common timeline (0 CE to 2010 CE, at yearly resolution) using cubic interpolation. Values were not extrapolated, so where the rebased years fall outside the original dataset, the record value was set as missing.



6. Metadata and the interpolated dataset were used to create two new CSV files.

These two files were loaded into the deep learning models, where additional filtering on location, timespan, and temporal resolution could be done quickly. We only used datasets that were located between 40–120°E and 10°S–50°N, had a decadal (or better) temporal resolution, and spanned at least 1500 CE to 1995 CE. After this filtering, there were 157 datasets that could be used to train the models. (Fig. 1).

## 2.2 Rainfall data

### 2.2.1 IMD

The Indian Meteorological Department (IMD) produces a daily 0.25°×0.25° gridded rainfall dataset for the whole of India (Pai et al., 2014). The dataset covers 1901 to present, and uses around 7000 rain gauges in total. The minimum amount of gauges used on any given day (early 1901) is about 1500. The gauge data are gridded using a simple inverse distance method (Shepard, 1968) adjusted to include directional effects and barriers (Rajeevan et al., 2006). The IMD dataset compares well against other overlapping datasets (Pai et al., 2014), except in the far north (mountainous Ladakh and Jammu and Kashmir) and far northeast (mountainous Arunachal Pradesh), where terrain makes gauges sparse and unreliable. We aggregated these data to create a training dataset containing the average rainfall during each monsoon season (June to September) for each gridpoint over India.

### 2.2.2 ERA5

We used precipitation data over South Asia derived from the ERA5 reanalysis data set (Hersbach et al., 2020). ERA5 provides hourly data at a horizontal resolution of 0.25°×0.25° from 1940 to the present. We aggregated these data to create a secondary training dataset containing the average rainfall during each monsoon season (June to September) for each gridpoint over India. ERA5 data are available from https://cds.climate.copernicus.eu/cdsapp#!/home).

### 2.2.3 Extended homogeneous region timelines

We used the datasets described in Sontakke et al. (2008) for our regional rainfall targets. These provide estimates of summer monsoon rainfall for seven 'homogeneous' zones in India (Fig. 1): North Mountainous India (NMI), Northwest India (NWI), North Central India (NCI), Northeast India (NEI), West Peninsular India (WPI), East Peninsular India (EPI), South Peninsular India (SPI), as well as averaged over the whole country. The datasets draw on slightly over 300 rain gauges spread across India. The datasets for each region have different start dates depending on the availability of records, ranging from 1813 (SPI) to 1848 (EPI). The authors did not assess the reliability of their data, but note that two regions, NEI and NMI, were very sparsely populated with gauges.



## 2.3 Famine data

Famine data were drawn from two aggregated lists: https://en.wikipedia.org/wiki/Timeline_of_major_famines_in_India_prior_
to_1765 and https://en.wikipedia.org/wiki/Timeline_of_major_famines_in_India_during_British_rule. In each case, the year(s),
affected region(s), and mortality (if known) are given for each famine, along with a list of supporting references. We convert
these to a table recording whether a famine occurred in each of the seven regions, or part thereof, for each year from 1500 to
1945. These data are available at https://doi.org/10.5281/zenodo.12688184. Given the complex relationship between famine
and drought, and the inconsistencies in recording historical famines, we only used famine data for illustrative purposes.

# 3    Methods

## 3.1    Bagging

Bagging, or bootstrap aggregation, is a way to reduce the risk of model overfitting. Neural networks trained on small datasets
have a high risk of overfitting. Because bagging promotes model diversity, as each model in the ensemble is trained with a
different dataset, the errors are at least partially orthogonal and, therefore, the overall variance is reduced by averaging out the
individual errors of each model in the ensemble.

Bagging begins with bootstrap sampling, where the convention is to draw multiple bootstrap samples from the original
training dataset with replacement, meaning the same instance can be selected multiple times in each bootstrap sample. This
procedure is not suitable for very small training datasets. Therefore, to keep the bias as low as possible, the bootstrap samples
for the timeline models are constructed by removing a random ten-year period for testing and another random five-year period
for validation. For the CNN models, each bootstrap sample has just a single test year and four validation years.

Independent models were then trained on each bootstrap sample. Because the bootstrap samples have random years taken out
for testing and validation, each model sees a slightly different version of the data. This process results in a diverse ensemble of
models, in a process similar to cross-validation. Once all the models are trained, bagging makes predictions by aggregating the
individual predictions of all the models in the ensemble. For regression problems, this typically involves taking some average
of the predictions of all models. In the timeline ensemble, we used the multi-model median, with the ensemble spread giving a
measure of the uncertainty.

## 3.2    Regularisation

We used $L_1$ regularisation, which introduces a penalty on the absolute values of the model parameters, thus encouraging
sparsity in the model's feature representations. This sparsity means that the model will learn to focus on a smaller subset of
features, effectively ignoring less relevant features. This is beneficial in reducing overfitting because it encourages the model
to ignore palaeoenvironmental records which do not reliably improve the predictions.



The $L_1$ regularisation term is given by

$$\lambda \Sigma |w_i| \tag{1}$$

where $w_i$ are the model parameters (i.e., weights) and $\lambda$ is a tuneable hyperparameter.

### 3.3 Dropout

We also used dropout to prevent overfitting, a technique where a fraction of neurons in a layer are randomly deactivated during training, with a specified probability, $p$, which is a tuneable hyperparameter, at each training step. This random deactivation forces the network to learn more robust and generalisable features, since the network learns to distribute the representation across all neurons instead of relying on any specific set of neurons (Liu et al., 2023). The randomness introduced by dropout helps break the complex co-adaptations among neurons, which can be particularly problematic when the training data is limited (Brigato and Iocchi, 2021). Dropout mitigates the tendency for neurons to become overly specialised to work well together on the training data, which causes them to perform poorly on new, unseen data, by promoting a form of ensemble learning within the network, where each sub-network (obtained by dropping a different set of neurons) learns to operate independently.

### 3.4 Blended loss function

The conventional choice of loss function for most regression tasks is mean squared error (MSE). However, given the small training dataset and requirement for a strongly regularised model, using the MSE means the models tend to regress towards the mean of the target distribution (i.e., zero, as we use standardised rainfall anomalies). To avoid this undesirable local minimum, we adjust the loss function as follows:

$$\text{Loss} = \alpha \cdot \text{MSE} + \beta \cdot \text{VarDiff} + \gamma \cdot \text{CorrLoss}, \tag{2}$$

where $\alpha$, $\beta$, and $\gamma$ are tuneable hyperparameters. The variance difference, VarDiff is given by:

$$\text{VarDiff} = |\text{Var}(y_{\text{true}}) - \text{Var}(y_{\text{pred}})|. \tag{3}$$

This component encourages the model to match the variance of the target distribution, and thus penalises direct regression to the mean. The correlation loss, CorLoss, is simply taken as:

$$\text{CorrLoss} = 1 - \rho(y_{\text{true}}, y_{\text{pred}}), \tag{4}$$

where $\rho(y_{\text{true}}, y_{\text{pred}})$ is the correlation coefficient between the observed and predicted target distributions. In the timeline model, this is a Pearson correlation coefficient computed between the two timeseries over a given time period. In the CNN autoencoder model, this is a pattern correlation coefficient computed over the gridpoints with valid rainfall observations. This component further discourages regression to the mean by encouraging the model to maintain a strong correlation between output and target.





| Layer Type | Output Shape | Activation | Regularisation | Other Parameters |
|---|---|---|---|---|
| Dense | (L1_neurons,) | L1_activation | $L_1$ (L1_regularisation) | - |
| Dropout | - | - | - | Rate: dropout |
| Dense | (L2_neurons,) | Linear | - | - |
| Dense | (1,) | Linear | - | - |

**Table 1.** General form of the architecture of the timeline models. Grey cells indicate regularisation or dropout and yellow cells indicate the encoder part of the model. See text for definition of terms, and Table 2 for a list of the configurations used.

| Config Number | L1_neurons | L2_neurons | L1_activation | dropout | L1_regularisation |
|---|---|---|---|---|---|
| 1 | 100 | 20 | ReLU | 0.2 | 0.01 |
| 2 | 100 | 20 | ReLU | 0 | 0.05 |
| 3 | 100 | 20 | Linear | 0.2 | 0.01 |
| 4 | 20 | 5 | ReLU | 0.1 | 0.01 |
| 5 | 100 | 10 | Linear | 0.2 | 0.001 |
| 6 | 50 | 5 | Linear | 0.2 | 0.01 |
| 7 | 50 | 5 | ReLU | 0.2 | 0.01 |

**Table 2.** Hyperparameter configurations used for the timeline models. L1_neurons and L2_neurons are the sizes of the first and second layers respectively; L1_activation is the activation function used in the first layer; dropout is the fraction of neurons randomly dropped from the first layer during training; and L1_regularisation is the magnitude of the $L_1$ regularisation term added to the loss function.

Values for $\alpha$, $\beta$, and $\gamma$ were found using a simple hyperparameter grid search. For the timeline model, we use $\alpha = 1$, $\beta = 1$, and $\gamma = 2$. For the CNN autoencoder model, we use $\alpha = 1$, $\beta = 0.75$, and $\gamma = 0.25$. The choice of $\alpha$ is of course arbitrary,
since it is the ratios $\beta/\alpha$ and $\gamma/\alpha$ that set the relative importance of the different components in the loss function.

### 3.5 Timeline model ensemble

We used two models to reconstruct monsoon rainfall over the last 500 years. The timeline model was used to produce time series of seasonal rainfall (averaged over each June-September) for each of the homogeneous rainfall regions and for the whole of India.

The architecture is simple (Table 1) – a dense feedforward neural network with two hidden layers separated by a dropout layer – and extremely cheap to run given the small training dataset. This allowed us to expand the bagged ensemble to include seven different hyperparameter configurations (Table 2), which were identified by a cross-validated grid search.

The full recipe for training and running these timeline models is given below. The process was repeated for each regionally-averaged rainfall dataset used.

1. Data preparation:





    (a) Define the geographical region (Fig. 1) and resolution (at least decadal frequency) for the palaeo-data.

    (b) Load and prepare the palaeo-data with the defined region and resolution.

    (c) Load the precipitation data for the relevant homogeneous region.

2. Model configuration:

    (a) Define a set of test and validation periods.

        i. For each hyperparameter configuration, create a set of 42 test periods – each a list of ten consecutive years starting on each of the even-numbered years from 1902 to 1984 inclusive.

        ii. Choose two validation years randomly from the remaining years, using a fixed seed to ensure repeatability.

    (b) Specify the model hyperparameters (number of neurons, activation functions, dropout rates, and regularisation; see Tab. 2).

3. Model training and evaluation:

    (a) For each combination of test period and hyperparameter configuration:

        i. Split the data into training, validation, and test sets.

        ii. Build and compile the model using the specified hyperparameters and architecture.

        iii. Train the model with the training data. At the end of each epoch, compute the loss (Sec. 3.4) on the validation data, and stop training if this has not improved over the last 250 epochs (Fig. 2).

        iv. Evaluate the model performance on the test set using Kling-Gupta Efficiency (KGE) and linear correlation coefficient. Log these scores in a file, along with a unique model identifier and metadata.

        v. Save the model weights using the same model identifier.

4. Model selection and prediction:

    (a) Select models from the log file that meet the desired performance criteria (i.e., KGE>0 and correlation coefficient>0.3).

    (b) Generate predictions for the extended period (1500–beginning of training dataset). Save these predictions to a file.

5. Ensemble and result compilation:

    (a) Load the results for all selected models and average them to create an bagged ensemble average.

For each ensemble member in each region, certain years within the training dataset were withheld for both validation and testing (see Sec. 3.5). The ensemble for each regional model then comprises those members whose validation $r > 0.5$, or the ten best-performing members, whichever yielded the greater number.





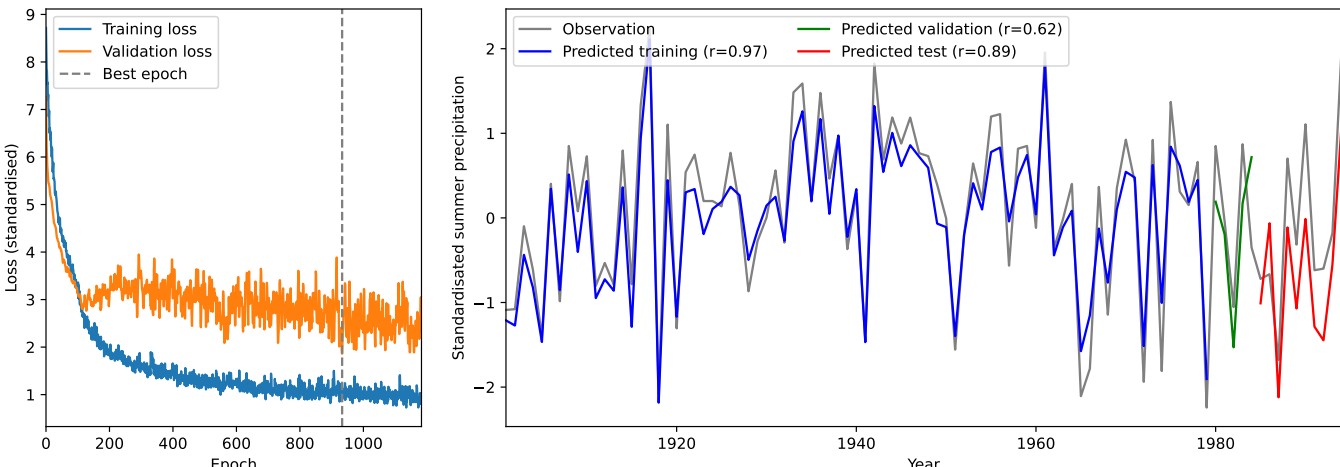

**Figure 2.** Example verification of one the timeline model ensemble members, trained on IMD gridded gauge precipitation averaged and standardised over the monsoon core zone. Left: training (blue) and validation (orange) loss curves as a function of epoch number. The dashed grey line indicates the best epoch, as determined by validation loss, from which the model weights are taken. Right: verification of the trained model against observations (grey) over the training period (blue; 1901–1979), verification period (green; 1980–1984) and testing period (red; 1985–1994). Correlation coefficients between model prediction and observation for each period is given in the legend.

## 3.6   CNN autoencoder ensemble

The creation of maps of rainfall anomalies, rather than a single area-averaged value, is a more complex task and therefore
requires more complex network architecture — convolutional layers within a convolutional neural network (CNN). Convolutional layers apply a set of learnable filters to the input data, where each filter slides (convolves) across the input to produce feature maps that capture various patterns or features. The resulting feature maps are then passed through activation functions to introduce non-linearity and enable the network to learn complex representations of the input data and then perform, e.g., classification tasks. They thus reduce the dimensionality of the input, usually in a way that results in a useful latent space,
e.g., detecting phases of ENSO from satellite images. Here, however, we want the network to decode rather than encode by taking low-dimensional palaeo-data and extracting a high-dimensional representation from it (i.e. rainfall maps). We thus invoke the deconvolution (or transposed convolution, Conv2DTranspose in Tab. 3). This expands input features to a larger spatial resolution. Like the convolutional layer, it achieves this by using a set of learnable filters (whose size, or 'kernel', must be pre-defined) that slide over the input data but simultaneously upscales by padding or striding.
The full CNN architecture (Tab. 3) comprises two parts: firstly, dense encoding layers transform the input palaeodata into a latent space, then a stack of deconvolutional layers decode the latent space into a rainfall anomaly map. A final cropping layer is required to adjust the size of the output to match the dimensions of the IMD training dataset (135 longitude × 129 latitude). As this model contains both encoding and decoding elements, it is referred to as an autoencoder.





Although the CNN is substantially more complex than the timeline model, the training and running is almost identical. The only differences are as follows.

1. The increased training time arising from greater model complexity means only one set of hyperparameters is used. These values are given in Tab. 3.

2. The choice of testing and validation years reflects both the increased model complexity and the shorter training dataset. Now, each model has one validation year and five test years. Every year (1901–1994) is used as a validation year four times. The associated testing years start 1, 3, 5, or 7 years after, and are then separated by twenty years, wrapping around if necessary. Thus the first model has 1901 as its validation year, and 1902, 1922, 1942, and 1962 as its testing years.

3. As described in Sec. 3.4, the functional form of the loss function is the same, but the parameters vary slightly.

4. The criteria for a model being used in the ensemble are (1) having a pattern correlation coefficient greater than 0.3 and (2) having a variance difference (for the standardised anomaly) of less than 0.1. Both are computed using on the validation year.

Like the timeline models, the CNN model comprises a number of ensemble members, each withholding small but different sets of years for validation and testing. As the model never sees data from testing years, performance for these years should be indicative of performance outside of the training dataset, i.e. from 1500–1900.

Two simple regression models, commonly used in palaeoclimate studies, were used to compare with the timeline and CNN models. Specifically, we used a generic multivariate linear regression model, where regression is computed directly for each pixel in the training data, and a PCA-based approach, where the training dataset dimensionality is reduced by decomposing it into eight EOFs, with regression computed over their corresponding principle components (from which the predicted field is reconstructed through the original EOFs). The choice of eight EOFs was made through inspection. Although model performance was not particularly sensitive to this choice, using many more or many fewer leads to reduced skill.

We thus use four models: linear regression and PCA, as standard techniques, timeline models, as they are fast, and CNNs as they are robust.

## 3.7 Shapley value analysis

### 3.7.1 Overview

Shapley values are widely used to assess the significance or contribution of specific variables in a model by estimating the marginal contribution from a given predictor in forcing a prediction away from the distribution mean (Shapley, 1953; Roth, 1988; Lundberg and Lee, 2017). The sum of all the Shapley values for all predictors for a given prediction, $\hat{Y}$, therefore, is equal to the difference between the predicted value and the predictand mean, i.e., $\hat{Y} - E(Y)$. We used the shap Python package (https://github.com/slundberg/shap), whose 'KernelExplainer' function estimates Shapley values using a kernel method. This is particularly useful for models where other forms of Shapley value estimation are computationally infeasible due to their





| Layer Type | Output Shape | Activation | Regularisation | Other Parameters |
|---|---|---|---|---|
| Dense | (512,) | Linear | $L_2$ (0.01) | - |
| Dropout | - | - | - | Rate: 0.2 |
| Dense | (256,) | ReLU | - | - |
| Dropout | - | - | - | Rate: 0.2 |
| Dense | (65536,) | ReLU | - | - |
| Reshape | (32, 32, 64) | - | - | - |
| Conv2DTranspose | (64, 64, 64) | ReLU | - | Kernel: (3,3), Stride: (2,2) |
| Conv2DTranspose | (192, 192, 32) | ReLU | - | Kernel: (3,3), Stride: (3,3) |
| Conv2DTranspose | (192, 192, 1) | Linear | - | Kernel: (3,3) |
| Cropping2D | (129, 135, 1) | - | - | Cropping: ((0,63),(0,57)) |

**Table 3.** Architecture of the CNN autoencoder models. Grey cells indicate regularisation or dropout, yellow cells indicate the encoder part of the model, and blue cells indicate the decoder part of the model. See text for definition of terms.

complexity. It works by building a local linear model that approximates changes in the output of the main model when a feature value is altered. This local model uses a background dataset to integrate out selected features, allowing it to estimate the impact of including or not including a given feature and thus its marginal contribution..

### 3.7.2   Implementation

We used two separate implementations. The 'backward' method takes a model prediction and used Shapley analysis to compute

the contribution of each palaeoenvironmental record to that prediction. Averaged over all predictions, this provides an estimate of the relative importance of each palaeoenvironmental record within the model for making rainfall predictions at each point. We did this using the timeline models as, although they are less robust than the CNN, the aggregation of the results by region means they are less prone to noise and small-scale spatial errors. This method shows which palaeoclimate records are not being used by the models and thus which may be defective or otherwise not useful (e.g., tree growth occurring entirely outside of

the monsoon season). This information could be useful for future modelling efforts since it can determine *a posteriori* which palaeoclimate records to exclude. The 'forward' method identifies the mean impact of a given palaeoenvironmental record on the predictions of each pixel in the CNN ensemble mean. This provides insights into how the model is using the palaeoclimate records, and thus how the model differs from and improves upon conventional multivariate linear regression or PCA techniques.







**Figure 3.** Estimated seasonal precipitation anomalies from the regional timeline model ensembles. For each of the seven homogeneous regions as well as all India (AI), the ensemble median is given in black and the spread in red. Observed values, taken from the reconstructed timeseries in Sontakke et al. (2008), are given in green. Where standardised anomalies lower than $-0.5$ occur in either the modelled or observed timeseries co-occur with known regional or national famines, these are marked with grey bands. Stated $r$-values measure the correlation between coincident actual and model test values.

# 4 Results

## 4.1 Timeline model

Five of the eight ensemble means had $r > 0.5$ over their respective test periods, with only one having $r < 0.4$. These values, while not exceptional, are better than some inter-observational agreement measures (Baudouin et al., 2020). Many of the reconstructed intervals of low rainfall coincide with famines (Fig. 3). The worst individual drought-famine year was 1630, in the middle of a multidecadal drought that persisted over the subcontinent throughout much of the mid 17th century (see, e.g.,





Singhvi and Kale, 2010). However, the seven regional timeline models (Fig. 3) tend to underestimate rainfall compared to the instrumental record, as does the All-India model. This seems to be an issue of timescale: the palaeoenvironmental records themselves capture decadal and multidecadal variability well, but strongly underestimate interannual variability. However, although the records are not necessarily good matches for monsoonal rainfall and there is poor coverage over much of the peninsula, the models generally do reasonably well. The size of the available training data has little impact on the model skill.

The worst performing region, SPI, has the longest training dataset, and the best performing region, EPI, one of the shortest. There is also no clear relationship between the number of records in a region and model skill. This suggests that the problem lies in the quality or usefulness of the records from each region (see Sec. 4.3). The fact that the timeline model method works reasonably well suggests that the palaeo-climate records contain sufficient information to reconstruct regional monsoon anomalies using the more complex CNN model to take advantage of the spatial relationships within the rainfall field itself

(Sec. 4.2).

## 4.2 CNN model

### 4.2.1 Evaluation

The five years (1906, 1926, 1946, 1966, 1986) used to test the single CNN ensemble member (Fig. 4) represent a range of conditions, with two deficient monsoons in 1966 and 1988 and three surplus monsoons in 1906, 1926, and 1946; each with

their largest anomalies in different regions. In 1906, the CNN model captures the dry anomaly over Jammu and Kashmir. However, while it captures the heterogeneous anomaly pattern over the rest of subcontinent, and hence accurately estimates the all-India average, it tends to locate these smaller-scale anomalies incorrectly, leading to a poor pattern correlation coefficient ($r = 0.07$). The other four years are better represented, with pattern correlation coefficients of 0.45 to 0.52. The CNN model accurately captures the anomalously wet monsoon troughs in 1926 and 1946 with dry conditions elsewhere, as well as the

anomalously dry conditions across most of the peninsula in 1966 and 1986. The locations of the large-scale anomalies in 1946 and 1986 are very well predicted (central India and west India respectively). In 1966, the dry anomaly is predicted to the south of central India as opposed to east of central India in the observations. In 1926, the wet anomaly is located over east central India, as opposed to the whole monsoon core zone in the observations.

The simple linear regression model (Fig. 4(c)) performs much less well than the CNN model. Only one year has a pattern

correlation coefficient close to the best years of the CNN model (r=0.45); the range for the other four years is 0.04-0.35. The linear regression model generally captures the observed wet and dry signals, with notable successes in 1926 (wet in central India, dry in the north and south) and 1986 (very dry in west India). However, the locations of the largest anomalies are typically wrong, e.g., it captures the surplus all-India rainfall in 1946 but overestimates it and places the strongest wet anomaly in far west India rather than over the monsoon trough. Similarly, it captures the deficient all-India rainfall in 1966, but makes west

(rather than east) India very dry. Thus, the linear regression model generally performs less well than the CNN model. Although PCA regression models are commonly used for spatial palaeoclimate reconstructions, it represents only a marginal (though





**Figure 4.** Example verification for one of the members of the CNN-IMD ensemble. (a) observed seasonal rainfall anomalies for the five years that were excluded from the model training dataset (computed using the IMD gridded gauge dataset). (b) CNN-IMD model predictions for the same five years, from one ensemble member. (c) predictions from a simple regression model, with the value of each pixel computed using multivariate linear regression (again excluding these five years from fitting). (d) predictions from a more complex regression model, computed by regressing eight PCAs onto the palaeoclimate data and then reconstructing the anomaly field from the related EOFs. The pattern correlation coefficient with observations is given in the top left for each model-year. In all cases, the correlation is computed using data coarsened to 2° resolution using a 9×9 median filter.





consistent) improvement on the linear regression model (Fig. 4(d)). The sign of the all-India rainfall anomaly is correct but, like the linear regression model, the anomalies occur in the wrong place.

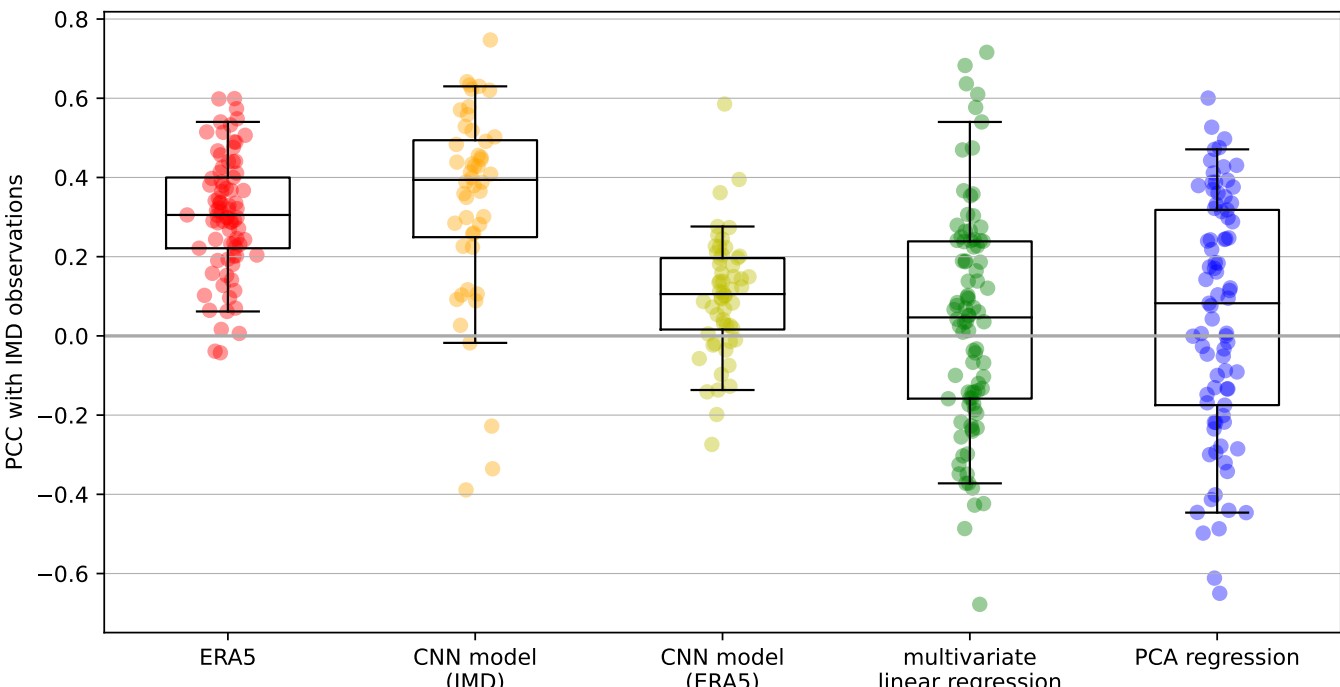

**Figure 5.** Distribution of pattern correlation coefficients between IMD observations and each model (plus ERA5 reanalysis) with each marker indicating the PCC for one monsoon season. The number of markers for each model is the number of years of overlap between the model and observations (1940–2022 for ERA5, 1901–1994 for CNN-IMD and the regression models, and 1940–1994 for CNN-ERA5. For the CNN models, the prediction from with the PCC is computed is taken as an ensemble mean of members that were not trained on the year in question. Similarly for the regression, one model was trained per five years of desired output, with the model not seeing those five year. As in Fig. 4, the PCC is computed using data coarsened to 2° resolution using a 9×9 median filter. The boxes represent the median and interquartile range, and the whiskers the 5th and 95th percentiles.

As a further test of the CNN model, we use the whole training dataset via cross-validation (Fig. 5) and add a CNN trained
on precipitation data from ERA5 (hereafter ERA5-CNN) compared to the original model (hereafter IMD-CNN). We used precipitation data from the ERA5 reanalysis to encompass the uncertainties due to differences between independent observational datasets. For each model, each point in Fig. 5 represents the pattern correlation coefficient for one monsoon season from the subset of members (or for regression, single model) not including that season in their training dataset. The IMD-CNN model is a closer match to the IMD observations than the ERA5 reanalysis, i.e., the error associated with this model is less than the
difference between the training dataset and another observational dataset. The median pattern correlation coefficient for the IMD-CNN is about 0.4, compared with about 0.3 for the ERA5 reanalysis. The ERA5-CNN is much weaker, with a median of about 0.1, partly because of the much shorter training dataset and partly because of the differences between the ERA5 training




dataset and the IMD evaluation dataset. The two regression models perform similarly to the ERA5-CNN, with median pattern correlation coefficients of about 0.1, though they have a much greater spread than either of the CNN models, with some years exceeding 0.5.


Comparison against June–September rainfall aggregated from the reconstructed monthly regional timeseries from Sontakke et al. (2008) provides a test for the IMD-CNN model for cases completely outside the IMD gridded gauge dataset. We use three years (1895, 1897, 1899; Fig. 6) that reflect diverse conditions: 1895 was an average monsoon with regional anomalies of both signs in peninsular India; 1897 was a surplus monsoon with all but NEI recording positive rainfall anomalies; and 1899 was a very deficient monsoon with four of the seven regions recording standardised rainfall anomalies of less than $-1.5$.


The IMD-CNN model (Fig. 6(b)) captures these variations well, especially the wet anomalies in the south in 1897 and the dry anomalies covering all of west and south India in 1899. It also captures the normal monsoon of 1895, although there are some errors with regional anomalies, especially in EPI. The IMD-CNN model generally has weak performance in NEI and NMI, likely due to relatively sparse observations in these regions in the training dataset, but potentially also due to large uncertainties in these regions in the Sontakke et al. (2008) dataset. In general, as is the case for the single ensemble model (Fig. 4), the IMD-CNN model tends to underestimate the magnitude of the anomalies.


The PCA regression model (Fig. 6(c)) captures the strong and weak monsoons of 1897 and 1899 respectively, including the general locations of the dry anomalies in west India in 1899. However, the model gets the locations of the wet anomalies wrong in 1897, and it predicts widespread wet anomalies in 1895 that were not observed.


The reconstructed Palmer drought severity index (PDSI) from Cook et al. (2010) is not expected to be a strict match (Fig. 6(d)), since the PDSI includes information on potential evapotranspiration. The PDSI is widely used in palaeoclimate studies because the inclusion of potential evapotranspiration means that it is often better correlated with tree ring records than rainfall alone. The PDSI correlates reasonably well with the instrumental record in 1895 and 1899, though, like the other PCA regression model, the anomalies are often in the wrong locations. In 1897, the strongly negative PDSI does not match the

observed slightly surplus monsoon. This may be due to lasting subsurface effects of a very weak 1896 monsoon.

The evaluation of the CNN model shows it performs better than other models, captures signals present in longer instrumental records, and is closer to the training dataset than a state-of-the-art reanalysis. This provides confidence in the use of the CNN model to make full reconstructions.

### 4.2.2 Reconstruction

We first examine the spatial patterns of extreme years, as shown by the CNN model and observations (Fig. 7). The four driest monsoons in the model – 1758, 1666, 1865, 1778 – all tend to have their strongest anomalies over western and southern peninsular India. This is broadly consistent with observations, indicating that the model has no consistent bias and is replicating patterns seen in the observations. In contrast, the strongest anomalies in the wettest years in the model occur in several different locations – from the far northwest in 1716 to the southern and central peninsula in 1731. Again, this pattern is consistent with

the wettest observed monsoons (compare, e.g., 1917 and 1988). Both results are consistent with theory: weak monsoons are characterised by long periods of low rainfall caused by large-scale intraseasonal circulation patterns (e.g., breaks). In contrast,



**Figure 6.** Verification of selected models against values for homogeneous regions given in Sontakke et al. (2008). The three years shown here (1895, 1897, 1899) are three of the last before the start of the training dataset. (a) reconstructed regional rainfall anomalies from instrumental records; (b) predicted values from the IMD-CNN ensemble mean; (c) predicted values from the PCA regression model; (d) values of PDSI from Cook et al. (2010).



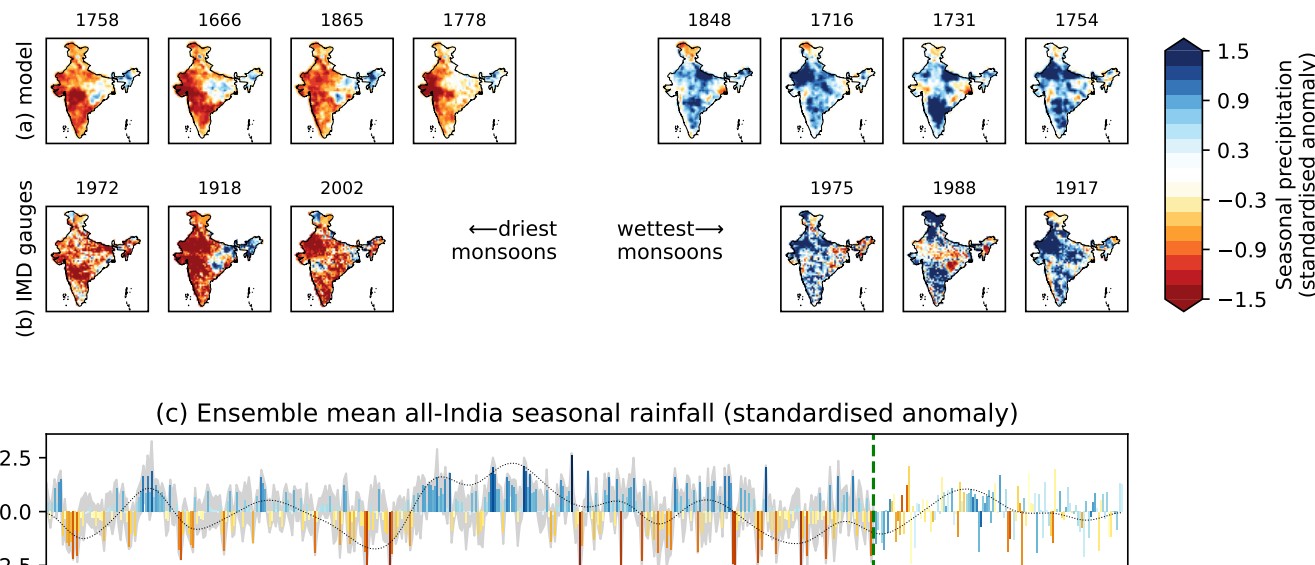

**Figure 7.** Maps of seasonal rainfall anomalies for the wettest and driest monsoon seasons in (a) the CNN ensemble mean (prior to 1901) and (b) the IMD gridded gauge observations. (c) All-India monsoon rainfall anomalies, computed using the CNN ensemble mean, with the IMD instrumental record to the right of the green lines. The dotted black line is a 10-year Gaussian low-pass filter (multiplied by a factor of 3). The grey shading in (c) indicates the bounds of the 11-member ensemble. In (a) and (b), the population of monsoons is sorted by total seasonal rainfall, i.e., by first converting the anomaly to an absolute amount using the spatial mean and variance computed from observations, thus reversing a preprocessing step before model training.

extreme wet anomalies are typically caused by the passage of one or several monsoon low pressure systems which, depending on their track, can cause heavy rainfall over most of India (Hunt and Fletcher, 2019; Thomas et al., 2021).

The model consistently underpredicts the magnitude of the anomalies, likely because it tends to overestimate the spatial scales of precipitation. Where observed extreme anomalies are marked by a strong heterogeneity and have high variance at small spatial scales (as well as large ones), this is not the case in the model. This is most likely due to the number of degrees of freedom in the predictor dataset being much less than in a high-resolution gridded rainfall product.

Nevertheless, since the large-scale signal is reasonable, we can extract the all-India rainfall timeseries (Fig. 7c) by converting the spatial anomalies to absolute values, taking a spatial mean and re-standardising the values for consistency. There is a marked
improvement in capturing the interannual variability compared to the original timeline model (Fig. 3), suggesting the model has successfully learned important spatial relationships in the training data, but it still somewhat underestimates the observed interannual variability (Fig. 7c). The decadal variability and timing of droughts during this period is corroborated by earlier work (e.g., Kathayat et al., 2022).

Consistent with the timeline model and other studies, conditions were generally dry in the seventeenth century (particularly
1650–1680), the eighteenth century was generally wet (particularly 1715–1740), and the nineteenth century had much stronger





interannual variability. The dry period in the middle of the nineteenth century is consistent with long-term gauge records (Sontakke and Singh, 1996). Quite surprisingly, given the general underestimation of interannual variability, the driest (1758) and wettest (1754) years predicted by the model are only four years apart. The ten wettest years are: 1551, 1715, 1716, 1731, 1732, 1754, 1762, 1848, 1917, and 1988. The ten driest years are: 1655, 1666, 1758, 1778, 1832, 1833, 1844, 1865, 1918, and

445   1972.

## 4.3   Explainability

Shapley analysis (Sec. 3.7), which shows the contribution of each palaeo-record to the prediction of the regional timeline models, indicates that the most important records tend to lie in the region of interest (Fig. 8). This is especially true when there is a large number of suitable, high quality palaeoclimate records. For example, in the regions bordering the Himalaya

(NMI, NCI, and NEI), where there are several long tree ring datasets, there are many records with high Shapley values. In contrast, records from regions with very few palaeoclimate records do not necessarily show good predictability for that region although they may be important for other regions. An example of this is the single speleothem record from EPI, which shows poor predictability for EPI though it is a useful predictor for the neighbouring WPI region. The lack of predictive power may be because the record does not have a strong causal relationship with regional monsoon rainfall or because the record is not

of high enough quality, e.g., because of low resolution. For the EPI speleothem, Shapley analysis implies the former case, and reflects the regional hydrology: at this latitude, several river basins stretch almost the whole width of the peninsula (e.g., the Godavari and Krishna rivers) and thus the heavier monsoon rainfall of the WPI may end up influencing speleothems in EPI.

Some palaeoclimate records are dynamically linked to rainfall in other regions. The two tree ring records in the Western Ghats, for example, are also useful for estimating rainfall over northern (NMI, NWI, NCI) and eastern (EPI) India. This

follows from the basic monsoon circulation: a stronger average monsoon is associated with both a stronger Somali jet, and hence enhanced moist flow and precipitation over the Western Ghats, and a stronger monsoon trough around which this moist flow subsequently recurves, bringing enhanced precipitation to the monsoon core zone, which covers much of NWI, NCI, and EPI. The records in the Arabian Sea and on the Somali coast are likely useful for the same reason. Records in the north can also be useful for modelling rainfall in the south, although this relationship is weaker due to the ratio of records in the two regions.

The model also ingests palaeoclimate records from outside India, most of which are from China. China experiences a summer monsoon (the East Asian monsoon) during roughly the same months as India. The two monsoons are linked both dynamically, via low pressure systems and wave responses to local diabatic heating and through shared teleconnections, e.g., with ENSO (Kripalani and Kulkarni, 2001; Shukla et al., 2011; Ha et al., 2018). The model is often able to use these records to constrain the estimates of regional precipitation over India, for example, the EPI model assigns high importance to a number of records

over south and central China.

The forward method indicates the approximate importance of specific records for predicting rainfall at each gridpoint in the CNN model ensemble mean predictions (Fig. 9). As an example, we use a typical central Himalayan tree-ring record (Fig. 9a). This dataset has a strong influence on local conditions along the Nepali border. However, it is also important for much of central India, especially along the monsoon core zone. This is likely due to the spatial coherence of seasonal anomalies in



**Figure 8.** Feature importance of each palaeoenvironmental record in each of the regional timeline model ensembles. Shapley analysis is applied to each of the first five models in each ensemble to determine the impact of each predictor on the final predictions. The points are coloured by the mean of the absolute values of these numbers.



**Figure 9.** Mean estimated Shapley value magnitudes for four of the input palaeoclimate records in the CNN ensemble model. The four records shown are: (a) a tree ring dataset in the central Himalayas in central Nepal; (b) a tree ring dataset at the southern edge of the Western Ghats in Kerala; (c) a tree ring dataset in the Hindu Kush in Ladakh; and (d) a speleothem from Hoq Cave on Socotra Island in the western Indian Ocean. The Shapley values are normalised such that a gridpoint value of 0.1 means that on average, the selected dataset changes the predicted value of standardised seasonally-averaged monsoon precipitation at that gridpoint by an average of 0.1.





this region: a strong monsoon trough, usually manifested through an increased frequency of low pressure systems crossing the region, results in increased precipitation both along the central Himalayan foothills and the Indo-Gangetic Plain.

This coherence, supported by synoptic-scale dynamics, is highlighted by one of the Keralan tree rings (Fig. 9b). It has a strong local impact, but through its link to the monsoon circulation it is also a useful predictor for much of north India. In particular, it seems to be useful at predicting extensions of the monsoon trough, with high impact to its west (Gujarat), north

(foothills), and south (Deccan Plateau).

However, many palaeoclimate records have only very localised impact in the model. This is, as expected from the results with the backward model, most common in areas with a high density of records. In such cases, the model finds that they provide only a small local adjustment. In the example of the Ladakhi tree-ring records (Fig. 9c), the model uses almost no information from this dataset outside of its immediate neighbourhood and even there it has a smaller impact than other nearby records.

The Hoq Cave speleothem on Socotra Island (Fig. 9d) is an example of an extra-regional record that has a big impact on model predictions of seasonal rainfall over much of central and west India because of the dynamically linkage between the two regions via the coupling of the Somali jet, which passes directly over Socotra, and the overall circulation strength of the summer monsoon. Precipitation over the two regions is correlated on interannual timescales (Fukushima et al., 2019), possibly driven by the IOD and ENSO (Jain et al., 2021), but there is also intraseasonal variability that affects both regions simultaneously,

independent of the low-level jet, such as the BSISO (Hunt and Turner, 2022).

### 4.4 The history of the monsoon

The reconstructions, based on both the timeseries (Fig. 3) and the CNN (Figs. 6 and 7) models, show considerable inter-annual variability in monsoon precipitation, particularly during the 19th and 20th centuries, but with marked decadal- and centennial-scale variability. during the earlier part of the record. The CNN model (Fig. 7) provides a somewhat smoother view of the

changes than the timeline model (Fig. 3) but there is good agreement between the two series. Across India as a whole, for example, there are intervals of below-average precipitation between 1507 and 1516 CE, with an extended period of drought from the end of the 16th century persisting through most of the 17th century, and with a further interval of below-average precipitation in the last decade of the 19th century. Comparison with the regional time series shows that each of these intervals affected different regions of the subcontinent. The earliest drought interval, for example, is registered quite strongly in WPI

and NMI, and less strongly in NCI, although individual years have below-average rainfall in many of the other regions of the country. The drought characteristic of the end of the 16th century and in the 17th century is registered more widely, including in EPI, WPI, NCI, NWI and NMI. Drought conditions at the end of the 17th century are strongly registered in EPI and also in NCI, NWI and NMI, although in these regions the anomalies are smaller though persistent over a longer period. There is a drought period between 1820 and 1840 CE which is strongly registered in EPI, but only shows up as individual anomalous

years in the all-India composite. Much of the 18th century is characterised by wetter-than-average conditions, and decadal-scale intervals of enhanced monsoons occur between 1542 and 1557 CE, 1681-1691 CE, 1700-1718 CE, and in the middle part of the 19th century.







**Figure 10.** Modelled monsoon rainfall anomalies for years associated with four major famines in the 17th and 18th centuries. In each case, the first recorded year of the famine is given in the middle row. 'Doji Bara' is the name given to a famine that affected much of south and west India.



The reconstructed dry extremes (Fig. 7) align well with known major famines (Fig. 10), with dry anomalies over the regions where famine was recorded. The "Entire India" famine of 1630 appears to have been triggered by widespread dry anomalies

reaching from the far south to the far north, with only Bengal and northeast India having a normal monsoon. Famines were recorded in Gujarat, Punjab, and Tamil Nadu in 1758, and the model predicts strong dry anomalies in these regions in that year, as well as across much of north India. The "Doji Bara" or "skull famine", which started in 1791, was triggered in part by a string of strong El Niños towards the end of the eighteenth century (Grove, 2007). This led to weak monsoons in 1788 and 1789, affecting west India and south India respectively, followed by a weak monsoon in southern India in 1790, and an

extremely weak monsoon in west India (with low rainfall across most of the peninsula) in 1791. This extended event led to catastrophic famine in Gujarat, Punjab, and Tamil Nadu, consistent with very large dry anomalies in the model. Our results for the spatial pattern of this drought are in agreement with Kathayat et al. (2022). In contrast to the strong negative anomalies of these three famines, the famine of 1711, which affected south India, Calcutta, and much of the east coast, was linked to two successive years of only mildly reduced rainfall in the these locations. In the cases where famines coincided with periods

of deficient monsoon rainfall, the strongest dry anomalies were typically co-located with the regions where the famines were recorded. These were often linked with consecutive years of negative regional anomalies, although this may be an artefact of the underestimation of interannual variability in the model.

## 5 Discussion

We have demonstrated the potential of deep learning techniques to produce robust spatiotemporal patterns of palaeomonsoon

rainfall over the Indian subcontinent. The reconstructions indicate that the high inter-annual variability characteristic of much of the 19th and 20th centuries is not typical of earlier periods, which displayed more decadal- to centennial-scale variability. Kathayat et al. (2022) have also drawn attention to this difference in the timescale of variability between the instrumental and pre-instrumental periods, based on a high-resolution speleothem record from Mawmulah. According to our reconstructions, the 17th century was drier- and much of the 18th century was wetter-than-average. Major droughts identified by the model,

for example during the early and late 17th century, correspond to documentation famines in the affected regions. Intervals of below-average rainfall are rarely registered across the whole of the continent, however, reflecting the fact that they are caused by different changes in circulation patterns and/or teleconnections. This suggests that the all-India monsoon index is, at best, an over-simplified way of contextualising changes in the monsoon. The spatial patterns produced by the CNN model provide a more nuanced way of examining patterns of change through time.

The spatial variability in the expression of above- or below-average monsoon precipitation also makes it difficult to rely on individual palaeo-records for reconstructions. Snow accumulation rates, and dust and chloride records from ice cores from the Tibetan Plateau and the Himalayas have been interpreted as indicators of changes in the monsoon over recent centuries Thompson et al. (2000); Davis and Thompson (2004); Yao and Yang (2004); Duan et al. (2004); Thompson et al. (2006), but there are large differences in the timing of inferred intervals of above- and below-average precipitation between different sites,

which are thought to reflect the impact of complex topography and potentially different precipitation sources (Kaspari et al.,





2008) which makes these records difficult to correlate with the overall behaviour of the monsoon across the peninsula. The high-resolution speleothem record from Mawmulah (not included in our training data) shows a relatively good concordance with our reconstructions of the timing of weak and strong monsoons over the past 500 years, but again this single record cannot be expected to capture all of the features shown by our reconstructions.

In our model reconstructions, as in many other palaeoclimate reconstructions, we make the assumption of stationarity. The CNN model learns from the period 1901–1994, and the timeline models extend this back a further fifty years or so. Both models assume that the seasonal relationships between palaeoclimate records and precipitation are fixed over this period, and that these relationships have held true since 1500 CE. This is an oversimplification. The monsoon has been subject to forcing from aerosols and greenhouse gases over this interval, both of which have a significant impact on seasonal precipitation (Lau

and Kim, 2017; Westervelt et al., 2020). Recent changes in rainfall seasonality, specifically a shift from winter rainfall to summer rainfall are thought to have affected the tree-ring records in the western Himalaya, for example (Kotlia et al., 2012; Munz et al., 2017). However, there is evidence that this variability has been present in the last century (Brunello et al., 2019; Fan et al., 2022) and is thus represented in the training dataset and should not present a problem for the models.The strength of teleconnections with the East Asian monsoon or ENSO is also not constant. The correlation between summer monsoon rainfall

and ENSO varies on decadal timescales (Torrence and Webster, 1999), and Rehfeld et al. (2013) showed that the coupling strength of the SAM and EASM varies centennially as a function of large-scale climatic conditions. However, the Shapley analysis indicates that the models does not depend heavily on these teleconnections.

The reliability of the training datasets will also affect the reliability of the final models. The gridded gauge dataset used as the rainfall target has very few gauges even today in the western and eastern Himalaya and the situation was even worse

in the first half of the twentieth century. This may explain why the CNN struggles to capture shorter wavelength signals. There may also be considerable dating uncertainties associated with the palaeoclimate records. Ljungqvist et al. (2016) assume an acceptable dating uncertainty of 200 years for the records in their dataset. Although most of the palaeoclimate records included in this study have an annual or near-annual sampling resolution this is not necessarily equivalent to the resolution of the climate signal. The transmission time for water transfer from the surface to the cave means that speleothem records,

for example, often have a signal resolution of several years (Baker et al., 2013; Comas-Bru et al., 2019). Similarly, annual tree growth can be affected by sub-optimal conditions in previous years such that the tree-ring record provides an attenuated signal of the current year (e.g., Moreau et al., 2020; Schnabel et al., 2022). Furthermore, although the monsoon is the dominant moisture source for tree growth over most of India, this is not necessarily the case in the western and central Himalaya where the covariance of annual and monsoon rainfall is moderate. However, our data-driven models have two advantages. Firstly, because

of the strong regularisation, they filter out predictors that have no tractable relationship with the target variable. Secondly, the models combine multiple palaeoclimate records non-linearly to extract local monsoon signals. Other studies have leveraged the dynamical links between the monsoon and more distant records to make reconstructions of changes in the monsoon (e.g., Burns et al., 2002; Fleitmann et al., 2007), but the advantage of data-driven models is being able to combine them with other records to maximise model skill at a given point. Thus, even if many of the palaeoclimate records used are not particularly





good indicators of monsoon rainfall and with relatively poor coverage over much of peninsular India, the models still have high skill.

## 6    Conclusions

We have used two types of deep learning model to produce palaeoclimate reconstructions of the Indian summer monsoon over the last 500 years. These were trained on long instrumental records of monsoon rainfall over India, and used records from 580    three palaeoenvironmental datasets covering South and East Asia. The small size of the training dataset necessitated strong regularisation, custom loss functions, and bagging. Both sets of models, therefore, were run as ensembles, from which we report the mean predictions.

    The timeline models were trained separately on seven regional rainfall datasets, ranging from 120 to 180 years in length, and their all-India sum. This produced reconstructions which captured decadal scale variability well, identifying the multidecadal 585    droughts of the mid seventeenth century which affected most of the subcontinent, as well as the dry conditions in the east and north of the country in the first half of the nineteenth century. However, despite reasonably high correlation with observations across their testing data (r∼0.5), these models largely underestimated interannual variability.

    The CNN model improves on these results by taking account of spatial relationships in the predictors and the rainfall. In testing, the CNN had a higher spatial correlation coefficient with the IMD observations, with an error smaller than that 590    between different observational datasets. The extreme deficit and extreme surplus years in the CNN model mirrored the spatial characteristics – in both scale and region – of the observations. The all-India monsoon timeseries produced by the CNN showed the following features:

- Multidecadal droughts from approximately 1640–1680 and approximately 1840–1900. The latter was characterised by much larger interannual variability.

– Multidecadal intervals of above-average rainfall from approximately 1690–1750.

- The most extreme drought years (1666, 1758, 1778, 1865) were characterised by very strong dry anomalies in south and west India.

- There were several events of large interannual variability (1754–58, 1776–78, 1784–88, 1828–33, 1844-48) where standardised seasonal rainfall of both signs exceeded a magnitude of 1.5.

– There were also several pairs of consecutive years where the seasonal anomaly fell below $-1.5$: 1654–55, 1666–67, 1758–59, and 1832–33.

Some of the identified drought years correspond to recorded famines; the dry anomalies typically occurred in those regions where the famine was reported, especially in the south and west.

    In the timeline model, the most important predictors in regions with ample high-quality palaeoclimate records tended to be 605    located in or near those regions due to their strong tie to the regional hydroclimate. However, in both the timeline and CNN



models, some records had predictive influence across broader regions, stemming from large-scale dynamical monsoon linkages. Some records in data-sparse regions had high importance if they related to the broader synoptic monsoon environment. This analysis indicates that the models had learned physically meaningful teleconnections between local palaeoclimate indicators and regional rainfall patterns.

*Data availability.* All data produced by this study are publicly available at https://doi.org/10.5281/zenodo.12688184.

*Author contributions.* KH conceived the study, designed and trained the model, and created the figures. SH and KH wrote and iterated the manuscript together.

*Competing interests.* The contact author has declared that none of the authors has any competing interests.

*Acknowledgements.* KMRH is supported by a NERC Independent Research Fellowship (MITRE; NE/W007924/1).



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
