# Peer review of "A novel explainable deep learning framework for reconstructing South Asian palaeomonsoons"

_EGUsphere, 2024_

## Author Comment (AC1)

The authors employed deep learning models to reconstruct paleomonsoon rainfall over India, using a range of paleoclimate records and two distinct approaches: one generating spatial map time series and the other producing regional time series. They implemented Convolutional Neural Networks for the spatial maps timeseries and multilayer perceptron models for the regional time series. The authors also incorporated an ensemble modelling strategy to enhance the robustness of their predictions. Although the predictive performance was not particularly strong, they effectively used the available dataset to perform data mining, and extracted as much information as possible, and I find that there methodological approach is pretty novel. Additionally, they applied explainable machine learning techniques to identify key predictors (paleoclimate records) across different locations, and supported their findings with physical knowledge.

The methods and results were well presented and thoroughly explained, with the authors fairly acknowledging the strengths and weaknesses of their findings and providing a well-rounded discussion. The datasets produced in the study have also been made available.

I recommend publishing this work, but I have a list of minor comments that could further improve it.

We would like to thank the reviewer for their positive assessment of our manuscript, and for their detailed comments, which we respond to point-by-point in red below. Textual revisions to our manuscript will be highlighted in blue.

**Specific comments**

1. When using the term "timeline models", readers might expect an architecture designed to handle time dependencies, such as recurrent neural networks or transformers. However, your MLP model does not inherently handle sequential or time-dependent data. I suggest using "regional models" to refer to MLPs, as you are applying them to predict region-based time series, and "spatial models" for the CNN models that produce spatial maps.
   This is a very good point regarding the timeline models – indeed the MLP cannot process temporal data and so the name is misleading. We have made the suggested change to "regional model" throughout. For consistency, we have also changed instances of "CNN model" to "spatial model", except where it is important to draw contrasts with other types of model (Section 4.2.1).

2. Line 33: ENSO acronym not defined
   We have fixed this.

3. Line 35: ITCZ acronym not defined, same for PDO in Line 59 and SST in Line 99
   We have fixed these.

4. Line 84: 'Machine learning approaches have been used in palaeoclimate research for automated palaeoenvironmental record generation, model post-processing'. Could you clarify what specific model is being referred to?
   This is a header statement for the paragraph that follows (in which we give many examples of these applications). We wanted to avoid giving too much specific ML-related jargon here as each case would then require an explanation for readers

unfamiliar with ML techniques. We thus use generic terms like "image detection". That said, there are some sentences in this paragraph where that description is too vague, and we have corrected them accordingly:

"Nelson et al. (2021) also used machine learning to improve and extend instrumental records." -> "Nelson et al. (2021) used a decision-tree based approach to improve and extend instrumental records".

"Machine learning has also been used to create backward models, for example, estimating tree ring width chronologies from local environmental factors (Jevšenak et al., 2018; Bodesheim et al., 2022; Li et al., 2023)." -> "Machine learning has also been used to create backward models, for example, estimating tree ring width chronologies from local environmental factors. These have used a range of techniques including multivariate linear regression (Jevšenak et al., 2018), decision trees (Bodesheim et al., 2022), and even deep learning (Li et al., 2023)."

Same applies to 'Machine learning has not been as widely used for model post-processing' in Line 93. Please specify which models you are discussing to enhance clarity.

As in the previous paragraph, we wanted to keep this quite generic – and so discuss the applications rather than particular methods. However, we have added some more detail here in our revision for the interested reader:

"Machine learning has not been as widely used for model post-processing, although it has been used to improve the temporal resolution of model output using frame interpolation methods (Zheng et al., 2024), to reconstruct output variables through nonlinear mappings (Huang et al., 2020), for anomaly detection (Bianchette et al., 2023), and for identifying droughts (Coats et al., 2020)." -> "Machine learning has not been as widely used for model post-processing, although it has been used to improve the temporal resolution of model output using frame interpolation methods (Zheng et al., 2024), to reconstruct output variables through nonlinear mappings (Huang et al., 2020), for anomaly detection using a multilayer perceptron (Bianchette et al., 2023), and for identifying droughts using Markov random fields (Coats et al., 2020)."

5. Line 98: There's a typo in byMalmgren
   Thank you, we have fixed this.

6. Line 99 and elsewhere: The term 'simple' MLP is not commonly used. Consider using a more standard term such as 'basic MLP' or just 'MLP'
   We have changed "simple" to "basic" throughout.

7. L100: There is an extra 'to'
   Thank you, we have fixed this.

8. Line 101: The description 'two-layer MLP' includes unnecessary detail. Consider simplifying it to just 'MLP'
   Agreed, we have removed this.

9. L116: Instead of 'to stabilize model training' it would be better to say: 'optimize the performance of the model'
   Agreed, we have made this change.

10. Figure 1: Are there blue contours present, or do you mean blue shades?
   Thank you for spotting this. We meant to say "filled blue contours" and have made that correction.

11. Section 2.1.4: This technical detail might be more suitable for the supplementary materials.
   Thank you for the suggestion. This is a very short subsection (~1/4 page) and it seems unnecessary to create supplementary material just to store it. We will defer this to the editor.

   Additionally, in point 6, please clarify the difference between the two CSV files.
   Sorry – our original wording here was unclear. We have revised this accordingly: "Metadata and the interpolated dataset were used to create two new CSV files." -> "Two new tabular files are created, one each for the metadata and the interpolated dataset."

12. Line 162: In 'After this filtering, there were 157 datasets that could be used to train the models. (Fig. 1),' please specify that these are the predictors of the models for clarity.
   We have made this change.

13. Line 203: how many years/samples were left for training?
   Thank you for pointing out this clarification. We have revised this paragraph as follows: "Therefore, to keep the bias as low as possible, the bootstrap samples for the regional models are constructed by removing a random ten-year period for testing and another random five-year period for validation. For the spatial models, each bootstrap sample has just a single test year and four validation years." -> "Therefore, to keep the bias as low as possible, the bootstrap samples for the regional models are constructed by removing a random ten-year period for testing and another random five-year period for validation. The number of training samples then varies depending on the region, with a maximum is 167 years and a minimum of 132. For the spatial models, each bootstrap sample has just a single test year and four validation years. This leaves 90 years of training data."

14. Line 204: These are not 'Independent models' since they share training data. It would be better to say 'separate models' or 'distinct models' instead.
   This is a good point. We have revised this to read "separate" in our revision.

15. Section 3.2 Regularisation: Instead of writing this as a separate section, consider combining it with the loss function section and include the equation of the loss function with the regularisation term.
   We agree and have made this change.

16. In Table 2, for clarity, update the caption to specify that L1 and L2 represent the hidden layers, and indicate the number of neurons in the input layer
   L1 actually denotes the input layer, with L2 denoting the hidden layer. We have made this clear in our revised caption.

17. Line 244: Was the choice of alpha arbitrary, or was it determined through trial and error?
   It is indeed arbitrary. As we state, what really matters for the revised loss function are the ratios beta/alpha and gamma/alpha. Two other ratios matter in training – if we were to double alpha (and beta and gamma), to achieve the same result we would also need to double lambda and halve the learning rate. However, as these values are determined

through the same hyperparameter optimisation, the choice of alpha can remain arbitrary. We have clarified this in our revision: "Values for α, β, γ were found using a simple hyperparameter grid search. For the regional model, we use α = 1, β = 1, and γ = 2. For the CNN autoencoder model, we use α = 1, β = 0.75, and γ = 0.25. For values of λ, see Sec. 3.4 and Sec. 3.5 for the regional and spatial models respectively. The choice of α is of course arbitrary, since it is the ratios β/α, γ/α, and λ/α that set the relative importance of the different components in the loss function. Note that to obtain the same results by doubling each of these parameters, we would need to halve the learning rate."

18. Lines 255-258: The 'Data Preparation' section is unnecessary. Instead, you could include details on how the predictors were standardised or normalised, as this information is currently missing.
We agree and have removed this subsection in our revision. As requested, we have added a new bullet point on data preprocessing: "Standardise the training data. Rescale each predictor in the training data to have a minimum of 0 and a maximum of 1 using min-max scaling. Then apply the same scaling parameters to the validation and test sets. For precipitation, normalise the training data to have a mean of 0 and a standard deviation of 1, and apply the same normalisation to the validation and test sets."

19. Figure 2: Where is the monsoon core zone?
This is leftover from a previous version of the manuscript and should read NCI (north central India). We have corrected this in the revision. We use the term later in our analysis, and now add a description: "In 1926, the wet anomaly is located over east central India, as opposed to the whole monsoon core zone (west, central and north India, excluding mountains) in the observations."

20. Lines 285-290: Instead of detailing how CNNs are typically used as encoders, focus on explaining your specific approach; starting with dense layers and then using CNN as a decoder, as mentioned in Line 290.
Thank you for the suggestion. Here, we wanted to give a brief overview of how CNNs work for readers who may not be familiar with them. We would prefer to retain this and will defer to the editor. However, we have added more detail on how transposed CNN layers work: "Specifically, transposed convolutional layers reverse the downsampling effect of standard convolutions by inserting zeros between input elements or adjusting strides, allowing the filters to produce outputs with larger spatial dimensions and reconstruct higher-resolution feature maps from lower-dimensional data." The next paragraph then begins by explaining the structure of the spatial network, starting with the dense layers.

21. Line 325: The term 'distribution mean' may not be accurate. It would be clearer to say 'the mean of all predictions.'
We have made this change.

22. In the caption of Figure 3: 'Observed values taken from the reconstructed time series in Sontakke et al. (2008) are given in green'. Do you mean the observed values are shown to the right of the green line?
Correct, this referred to a previous version of the figure. We have now fixed this.

23. Line 367: The term 'pattern correlation' is ambiguous as it could refer to either 'temporal' or 'spatial' correlation. Please specify that this refers to 'spatial' correlation for clarity.

Thank you for this suggestion. We have made this change here and throughout our revised manuscript.

24. Figure 5 caption: Please revise the sentence 'the prediction from with the PCC is computed is taken' for clarity.

Thank you – we have revised this to: "For the spatial models, the PCC is computed over the ensemble mean of members that were not trained on the year in question."

Additionally, ensure consistency between 'CNN-ERA5' in the caption and 'ERA5-CNN' mentioned in Line 385.

Thank you for spotting this, now corrected.

25. Figure 7 caption: There is no dotted black line as mentioned; instead, there is a single dotted green line. Please correct the caption to reflect that it is one dotted green line, not 'green lines'.

Thank you, we have fixed the caption to refer to the dashed green line (which again referred to an older version of this figure). Although the black line in (c) is dotted, this is not obvious in the manuscript version of the figure and so we replace "dotted" with "thin".

26. Line 455: Are you referring to low 'temporal' resolution?

Yes, we have clarified this in the revision.

27. Figure 8: How were the Shapley values standardised?

The "standardised" here refers to the fact that the precipitation data are standardised before training and are not subsequently converted back to their original values. Therefore, the Shapley values apply to this standardised dataset, which we now explain in our revised caption: "The Shapley values are standardised in the sense that a value of 0.1 means that on average, the selected dataset changes the predicted value of standardised seasonally-averaged monsoon precipitation in the given regional model by 0.1."

Also, did you take the absolute values of the Shapley values before averaging them? Please add these details for clarity.

Yes, that's correct. We've added this in the revised caption.

---

## Author Comment (AC2)

**General Comments**

In their manuscript Hunt and Harrison provide a novel approach to reconstructing historical monsoon variability: using machine learning to assess the relationship between paleoclimate records as predictors and the IMD gridded rainfall dataset between 1901-present as the target, and then extending that record back through time using annually resolved paleoclimate records from tree-rings, speleothems, glaciers etc. The results are interesting and appear to provide a significant step forwards monsoon predictability, comparable or slightly better than previous techniques in developing time series at point locations or areal means, but a significant advance in reconstructing spatial heterogeneity of monsoonal variability. The section on assessing the importance of individual paleoclimate records in the model, and the 'sphere of influence' of individual sites is an exciting development and appears to be backed by our understanding of monsoon dynamics.

The manuscript is well written, and the language is largely appropriate but could be simplified in places to the intended audience of Climate of The Past who likely have a more limited understanding of machine learning techniques. This includes more high-level introduction to methods, and care explaining predictors, targets, training datasets, validation datasets, test datasets etc. There is also sometimes a disconnect between what is labelled on the figures, the figure captions and what the figures refer to. Consistency and clarity are needed in places.

I do not have any significant concerns over the science and resulting inferences presented in this manuscript, though I will happily defer to more qualified machine learning reviewers on the robustness of the methods. I believe that this paper is suitable for publication in Climate in the Past, after correction for some minor issues.

Thank-you

Nick Scroxton

Maynooth University

We would like to thank Nick for his positive assessment of our manuscript, and for his comments, which we respond to point-by-point in red below. Textual revisions to our manuscript will be highlighted in blue.

**Specific Comments**

- Could you explain LiPDs in either section 2.1.1 or 2.1.2 before you get to line 148. Thank you for the suggestion. We have added the following at the end of Sec 2.1.3: "All data from this database, as well as from PAGES2k and Iso2k, are available as Linked Paleo Data (LiPD) files. These provide a wealth of metadata and a standard format that makes them machine-readable."

- Section 3: A high-level couple of sentences at the start of the methods would help non-machine learning experts. For example, it doesn't actually say in the manuscript what is your predictor data-set is and what is the target data-set. We agree it would be useful to have a general introduction at the beginning of the methods. We have added the following at the beginning of Section 3: "In this study, we aim to reconstruct historical monsoon rainfall over India over the last 500 year using

deep learning models. Our predictor dataset – i.e., the model input – comprises a wide range of palaeoenvironmental records (Sec. 2.1) interpolated to annual resolution. Our target datasets – i.e., what we want the model to predict – are derived from observed monsoon rainfall (Sec. 2.2) The models were trained and tested on replicating the target datasets over their lengths, and once testing confirmed their predictions were robust, their predictions were extended backwards in time to 1500. To achieve this reconstruction, we tested two different architectures of model. The first was a regional model, built using a dense multilayer perceptron, and was trained to replicate longer instrumental timeseries of precipitation over the homogeneous regions (Sec. 2.2.3). The second was a spatial model, built using a decoding convolutional neural network, trained to replicate gridded precipitation data (Sec. 2.2.1). To avoid overfitting on small training datasets, we employed a range of common techniques including bagging, dropout, and regularisation. Finally, to understand how the models make their predictions given certain inputs, we employed an explainability method known as Shapley analysis. These models and techniques are described in greater detail in the sections below."

- What's the difference between validation and test datasets in Figure 2 and section 3.5 1a. What do they do and why?
  This is already explained briefly in Sec 3.4 (part 2.a.iv of the recipe). For clarity, we have also added this information to Sec 3.1, where the term "validation" is first introduced, in our revision: "The purpose of having distinct validation and training datasets also arises from the desire to avoid overfitting. As the models are trained, the loss function is computed on both the training dataset (which the training process is designed to minimise) and the validation dataset. If the validation loss starts to increase while the training loss continues to decrease, that is a sign that the model is starting to overfit. However, because the model has thus been tuned on the validation data, a true fair test requires that it is distinct from the test dataset, which must remain hidden from the model."

- Figure 3: This figure does not show clearly the information that is attributed to it. There are significant mismatches between the figure and figure caption. I don't see the ensemble median in black, the spread in red, or the Sontakke time series in green.
  Thank you for noticing this. This caption referred to an earlier version of the figure. We have now updated it: "Estimated seasonal precipitation anomalies from the regional model ensembles. For each of the seven homogeneous regions as well as all India (AI), the ensemble median is given by the coloured bars. Observed values, taken from the reconstructed timeseries in Sontakke et al. (2008), are given to the right of the green line. Where standardised anomalies lower than −0.5 occur in either the modelled or observed timeseries co-occur with known regional or national famines, these are marked with grey bands. Stated $r$-values measure the correlation between coincident actual and model test values."

  I think this figure should be expanded to take-up more space on the page to really highlight the key results attributed to it.
  We agree and have made this change.

- Line 380-383: The dismissal of the PCA technique is too strong at this stage of the manuscript. At this point the reader has only been introduced to figure 4. The PCA outperforms the CNN model 40% of the time (2 out of 5 years) in figure 4 so cannot be dismissed, although an argument can be made that it lacks spatial heterogeneity. Once we have seen figure 5 and the larger dataset, we see that outperformance of the PCA method in figure 4 is likely just an artifact of small sample size, and therefore the dismissal is more reasonable. This section therefore might need to be reworded

  This is true. We have added the following at the end of this section: "However, this is just one ensemble member compared across just five seasons. While the mean value of $r$ is higher in the CNN (0.40) than the PCA (0.31) and linear (0.25) models, the latter two do beat the CNN model in two of the five years. Therefore, as a further test..."

- I wonder if dry years also correspond to major volcanic eruptions (as we might expect). This might provide an additional test of reconstruction performance that is less reliant on additional human societal complexities.

  Thank you for this very useful suggestion. We have created a new figure (Figure 8) which takes the area-averaged timeseries from what was Fig 7c and shows it alongside a popular ENSO reconstruction and historic large eruptions (VEI>4). We have added some new text describing the relationships that emerge: "The reconstructed ENSO anomalies (Fig. 8(b)) have a correlation with monsoon anomalies that varies centennially: the rolling 30-year correlation (not shown) is negative throughout almost all of the 18th and 20th centuries, but is positive in the 17th and early 19th centuries. This pattern is consistent with previous studies (Shi and Wang, 2019), in which it is speculated to be modulated by the Pacific Decadal Oscillation. Similarly, large volcanic eruptions (Fig. 8(c)) have a significant impact on the reconstructed monsoon anomalies. Lagged composites of monsoon rainfall for each VEI (not shown) suggest the impact on the monsoon grows with increasing VEI. In year 0 of VEI5 events, the mean reconstructed monsoon rainfall anomaly falls to −0.11, then to a minimum of −0.18 in year 2, before recovering by year 4. The pattern is different in VEI6 events, which initially cause an increase in monsoon strength, reaching a maximum mean anomaly of +0.25 in year 2. This then starts to fall and becomes negative by year 5, reaching a minimum of −0.56 in year 8, after which it recovers. These coherent responses of the reconstructed monsoon to volcanic eruptions gives us further confidence in our reconstruction."

- I disagree with some of your speleothem inferences. In section 4.3 If the EPI speleothem record is d18O then we might expect a better correlation with WPI rainfall than EPI, and thus this example belongs in the following paragraph instead. I don't see the relevance of the river basins argument here on speleothem proxy variability.

  We have not expressed this argument clearly. We agree that the river basin argument is not directly relevant. The EPI speleothem record is d18O from Jhumar Cave. Sinha et al. (2011) have shown that there is a significant inverse relationship between the changes in 18O and regional rainfall in the observational period and have argued that the variability in this record therefore reflects changes in both local and upstream rainfall. Thus, it is plausible that the record is influenced by the heavier monsoon rainfall of the WPI, as implied by the Shapley analysis. We have revised the relevant text so that it now reads: "An example of this is the single speleothem record from EPI (Jhumar Cave), which shows poor predictability for EPI though it is a useful predictor for the neighbouring WPI

region. The lack of predictive power may be because the record does not have a strong causal relationship with regional monsoon rainfall or because the record is not of high enough quality, e.g., because of low resolution. Shapley analysis implies the former case for the EPI speleothem. Sinha et al. (2011) have argued that the variability in this record reflects changes in both local and upstream rainfall and thus it is plausible that the heavier monsoon rainfall of the WPI may end up influencing speleothems in EPI."

**Technical Corrections**

- Line 196-198: Could you unreversed the first half of this sentence for clarity
  We have rewritten this sentence: "Bagging promotes diversity among models by training each one on a different dataset. This diversity causes the errors of each model to be at least partially orthogonal and as a result, averaging the errors across the ensemble reduces the overall variance."

- Figure 2 caption: 'one of the timeline model'
  Thank you, we have fixed this.

- Line 354: New paragraph?
  We have made this change.

- Line 364: 1988?
  This should read 1986. This has been corrected.

- Line 380: New paragraph?
  We have made this change.

- Line 528, 542: The cave name is spelt 'Mawmluh' or 'Krem Mawmluh" if cave needs to be included.
  Thank you for this suggestion. We originally followed the spelling given in Kathayat et al. (2022), but now understand this is nonstandard and have corrected our manuscript accordingly.

- Line 530: 'documented'?
  Thank you, we have made this change.

- Line 531: 'subcontinent' to be consistent throughout the manuscript.
  We have replaced "continent" here with "subcontinent". That was the only such instance we found in the manuscript.

- Line 561: clarify what you mean by wavelength or use less technical language.
  We have replaced "shorter wavelength signals" with "finer spatial detail".

- Line 578: delete 'have'
  Fixed.